# Stunting among children under two years in Indonesia: Does maternal education matter?

Agung Dwi Laksono[1,2], Ratna Dwi Wulandari[2,3]*, Nurillah Amaliah[4], Ratih Wirapuspita Wisnuwardani[5]

1 National Research and Innovation Agency, Republic of Indonesia, Jakarta, Indonesia, 2 The Airlangga Centre for Health Policy, Surabaya, Indonesia, 3 Faculty of Public Health, Universitas Airlangga, Surabaya, Indonesia, 4 Center for Research and Development of Public Health Efforts, Ministry of Health of The Republic of Indonesia, Jakarta, Indonesia, 5 Faculty of Public Health, Mulawarman University, Samarinda, Indonesia

* ratna-d-w@fkm.unair.ac.id

**Data Availability Statement:** The 2017 Nutrition Status Monitoring Survey data used to support these findings of this study were supplied by the Directorate of Community Nutrition of the Indonesian Ministry of Health under license and so

## Abstract

### Background

Measuring height for age is one of the essential indicators for evaluating children's growth. The study analyzes the association between maternal education and stunting among children under two years in Indonesia.

### Methods

The study employed secondary data from the 2017 Indonesia Nutritional Status Monitoring Survey. The unit of analysis was children under two years, and the study obtained weighted samples of 70,293 children. Besides maternal education, other independent variables analyzed in this study were residence, maternal age, maternal marital status, maternal employment, children's age, and gender. In the final stage, the study occupied a multivariate test by binary logistic regression test.

### Results

The results show the proportion of stunted children under two years in Indonesia nationally is 20.1%. Mothers in primary school and under education categories are 1.587 times more likely than mothers with a college education to have stunted children under two years (95% CI 1.576–1.598). Meanwhile, mothers with a junior high school education have a chance of 1.430 times more than mothers with a college education to have stunted children under two years (95% CI 1.420–1.440). Moreover, mothers with education in the senior high school category have 1.230 times more chances than mothers with a college education to have stunted children under two years (95% CI 1.222–1.238).

### Conclusion

The study concluded that the maternal education level was associated with stunting children under two years in Indonesia. The lower the mother's level of education, the higher the chances of a mother having stunted children under two years.

can not be made freely available. Requests for access to these data should be made to the Directorate of Community Nutrition of the Indonesian Ministry of Health (https://gizi.kemkes.go.id/; Email: subditkewaspadaangizi@gmail.com).

**Funding:** The authors received no specific funding for this work.

**Competing interests:** The authors have declared that no competing interests exist.

## Introduction

Stunting is when children under five years old (toddlers) have a length or height less than their age—the condition by a length or height of more than minus two standard deviations of WHO's median child growth standard [1–3].

Stunting is irreversible due to inadequate nutrition and repeated infections during the first 1000 days of a child's life. Childhood stunting is one of the most significant barriers to human development and globally affects an estimated 162 million children under five. Stunting has long-term associations on individuals and society, including decreased cognitive and physical development, reduced productive capacity and poor health, and an increased risk of degenerative diseases such as diabetes [2, 4]. Stunting is a well-established risk marker of poor child development, and Stunting before age two predicts poorer cognitive and educational outcomes in later childhood and adolescence. Child stunting has immediate and long-term consequences, including increased morbidity, mortality, and adverse impact on child development and adult health contributes to the cycle of malnutrition, and hampers economic development [3, 5].

The prevalence of short toddlers in Indonesia tends to be static. The 2007 Indonesian Basic Health Survey results showed the prevalence of stunting in Indonesia was 36.8%. In 2010, there was a slight decrease to 35.6%. However, most stunted toddlers increased again in 2013 to 37.2%. In 2018, the survey found that the prevalence of stunting in children under two years is 29.9%. In toddlers, it is 30.8% [6]. Moreover, the stunting prevalence in 2019 was 27.67% [7].

The incidence of stunting is still one of the nutritional problems experienced by toddlers in the world today. In 2017, 22.2% or around 150.8 million under-five experienced stunting. However, this figure has decreased compared to the stunting rate in 2000, which was 32.6%. In 2017, more than half of stunted children under five came from Asia (55%), while more than a third (39%) lived in Africa. Of the 83.6 million small children under five in Asia, the highest proportion came from South Asia (58.7%). Data on the prevalence of stunting under five collected by WHO, Indonesia is included in the third country with the highest majority in the Southeast Asia/South-East Asia Regional (SEAR) region. The average prevalence of stunting under five in Indonesia from 2005 to 2017 was 36.4% [8, 9].

The WHO Conceptual Framework on Childhood Stunting describes how stunting is caused by a complex combination of family, environmental, social, and cultural variables [10]. Stunting is a chronic nutritional problem (a condition that lasts a long time) caused by many factors such as socioeconomic conditions, maternal nutrition during pregnancy, and infant pain [11]. Other causes are unhealthy living behavior, and lack of food intake for a long time from infancy can cause children to become short [6].

One of the demographic characteristics that became the focus is maternal education. Education is a critical factor that does not directly affect nutritional status because this education will affect the pattern of parenting for children [12]. The study analyzed data on 85,932 children aged 6–59 months in Vietnam and found that there was no maternal education among children 6–23 months, compared with a graduate education (OR 1.77; 95% CI, 1.44–2.16). Meanwhile, for children 24–59 months, the strongest associations with child stunting were no maternal education compared with a graduate education (OR 2.07; 95% CI 1.79–2.40) [13].

Meanwhile, another study conducted in Indonesia with a cohort-prospective study yielded similar conclusions. The study performs between August 2012 and May 2014 at three health centers in Jakarta, Indonesia. Subjects were healthy children under two years old, in which the study measured their weight and height serially (at 6–11 weeks old and 18–24 months old). Of 160 subjects, 14 (8.7%) showed declined growth pattern from regular to stunted and 10 (6.2%)

to severely stunted. As many as 134 (83.8%) subjects showed consistent standard growth patterns. Only two (1.2%) showed improvement in linear growth. Maternal education duration of fewer than nine years (OR 2.60, 95% CI 1.23–5.46) showed a statistically significant association with declined linear growth in children. Mother with an education duration of fewer than nine years was the determining socio-demographic risk factor contributing to the decreased linear growth in children under two years of age [14]. Based on the description of the background narrative, we intend this study to analyze the association between maternal education level and the stunting among children under two years in Indonesia.

## Materials and methods

### Data source

The data used in this analysis is secondary data from the 2017 Indonesia Nutritional Status Monitoring Survey. The 2017 Indonesian Nutritional Status Monitoring is a cross-sectional survey on a national scale conducted by the Directorate of Nutrition of the Indonesian Ministry of Health. The design of this study is cross-sectional, which analyzes data of variables collected at one given time across a *sample* population. The aim is to collect and present systematic data to provide a factual description of a particular situation.

Sample determination of under five-year samples: Sample selected 300 households. Samples of children under five years are all children under five years in homes chosen in each cluster. Respondents are mothers of children under five years or household representatives who can represent the sample [15].

The 2017 Indonesian Nutritional Status Monitoring uses a minimum of a nutrition diploma graduate as interviewers and anthropometric measurements. Measure height using a microtoise, while body length using a length board—measurement of body weight with a digital scale with an accuracy of 0.01 kg.

The population in this study were all children under two years in Indonesia. In this study, the unit of analysis was children under two years (<23 months), with mothers as respondents. The weighted sample was 70,293 children under two years with the multi-stage cluster random sampling method.

### Variables

The study employed stunted children under two years as an outcome variable. Stunting was a nutritional status indicator based on height for age or the height of a child who is reached at a certain age. Based on WHO growth standards, the height indicator for a period is determined based on the z-score or height deviation from average height. Stunted children under two years consist of two categories: not stunted and stunting. The limit for the nutritional status category according to the height index/age is [15]:

- Stunted: $< -3.0$ SD to $-2.0$ SD

- Normal: $\geq -2.0$ SD

The research used maternal education as an exposure variable. The survey determines maternal education based on the last certificate held by mothers of children under two years. Maternal education consists of four levels: primary school and under, junior high school, senior high school, and college.

Apart from maternal education level, other independent variables, as control variables, were the type of residence, maternal age, maternal marital status, maternal employment status, age of children under two years, and gender of children under two years. The type of residence

consists of two types: urban and rural. Maternal age is determined based on the last birthday (in years). Maternal marital status includes never married, married, and widowed/divorced. Moreover, maternal employment status consists of two categories: unemployed and employed.

Children under two years are determined based on the last month's birthday (in months). Meanwhile, the gender of children under two years consists of two types: boy and girl.

The inclusion criteria in this study were children under two years. On the other hand, the exclusion criteria in this study were children under two years who did not have a mother, and their anthropometry was not measured.

### Data analysis

The study carried out a co-linearity test in the early stages of analysis. Then, the study used the Chi-Square test to test the dichotomous variables, and the T-test for continuous variables. The study used the statistical test to assess whether there is a statistically significant relationship between the variable nutritional status of children under two years as the dependent variable and the independent variable. The study uses a multivariable test in the final stage by utilizing a binary logistic regression test. The analysis performed all statistical analyzes with IBM SPSS Statistics 21 software.

Moreover, the research used ArcGIS 10.3 (ESRI Inc., Redlands, CA, USA) to create a distribution map of stunted children under two years in Indonesia. The study issued a shapefile of administrative boundary polygons by the Indonesian Bureau of Statistics for the task.

### Ethical approval

The 2017 Indonesian Nutritional Status Monitoring Survey has an ethical license approved by the national ethics committee (Number: LB.02.01/2/KE.244/2017). The survey used informed consent during data collection, which accounted for aspects of the procedure for data collection, voluntary and confidentiality. Respondents gave written consent.

## Results

The analysis results indicate that the proportion of nationally stunted children under two years in Indonesia is 20.1%. The lowest proportion of stunted children under two years was in Bali Province at 13.6%; meanwhile, the province with the highest proportion of stunted children under two years was Central Kalimantan Province at 30.1%.

Table 1 shows that co-linearity tests indicate no collinearity between independent variables. Based on Table 1, the analysis results show that the tolerance value for all variables is more significant than 0.10. At the same time, the variance inflation factor (VIF) value for all variables is less than 10.00. Then referring to the basis of decision-making in the multicollinearity test, the study concluded that there are no symptoms of a strong relationship between two or more independent variables in the regression model.

### Descriptive analysis

Table 2 shows a statistical description of the characteristics of children under two years who are the object of analysis in this study. The value of the proportion of children under two years living in rural areas in Indonesia is 22.6% (95% CI 22.4%-22.8%). Children under two years living in rural areas dominate all nutritional status categories based on the type of residence.

According to maternal education, mothers with senior high school education led in nutritional status categories. Based on maternal age, stunted children under two years have mothers with an average age slightly older than normal children.

**Table 1. The results for the co-linearity test of nutritional status of children under two years in Indonesia (n = 70,293).**

| Variables | Collinearity Statistics | |
|---|---|---|
| | Tolerance | VIF |
| **Area context** | | |
| Residence | 0.961 | 1.040 |
| **Maternal Characteristics** | | |
| Education level | 0.936 | 1.069 |
| Age (in years) | 0.980 | 1.021 |
| Marital status | 0.997 | 1.003 |
| Employment Status | 0.962 | 1.040 |
| **Children under two years' Characteristic** | | |
| Age | 0.996 | 1.004 |
| Gender | 1.000 | 1.000 |

Note: *Dependent Variable: Nutritional status of a toddler; VIF: variance inflation factor.

Based on maternal marital status, married mothers led both nutritional status categories. On the other hand, according to maternal employment status, unemployed mothers dominate in both types of nutritional status.

Table 2 shows that average children under two years who are stunting are older than children under two years who have normal nutritional status. Moreover, based on children under

**Table 2. Descriptive statistic of nutritional status of children under two years in Indonesia (n = 70,293).**

| Variables | Nutritional Status | | p-value |
|---|---|---|---|
| | Not stunted (n = 55,142) | Stunting (n = 15,152) | |
| **Residence** | | | < 0.001 |
| • Urban | 26.7% | 22.6% | |
| • Rural | 73.3% | 77.4% | |
| **Maternal Characteristics** | | | |
| Education level | | | < 0.001 |
| • Primary school and Under | 26.4% | 30.6% | |
| • Junior high school | 26.8% | 28.6% | |
| • Senior high school | 37.8% | 34.1% | |
| • College | 9.0% | 6.6% | |
| Age (in years; mean) | 29.63 | 29.72 | < 0.001 |
| Marital status | | | < 0.001 |
| • Never married | 0.3% | 0.4% | |
| • Married | 98.9% | 98.4% | |
| • Divorce/Widowed | 0.8% | 1.1% | |
| Employment Status | | | < 0.001 |
| • Unemployed | 76.5% | 76.4% | |
| • Employed | 23.5% | 23.6% | |
| **Children under two years' Characteristic** | | | |
| Age (in months; mean) | 10.67 | 15.0 | < 0.001 |
| Gender | | | < 0.001 |
| • Boy | 49.8% | 56.7% | |
| • Girl | 50.2% | 43.3% | |

two years of gender, the boy led in the stunted category; On the contrary, the girl dominated the not stunted type.

## Multivariate analysis

Table 3 shows the results of the binary regression logistics to analyze the association between maternal education level and the stunted among children under two years in Indonesia. The study used the nutritional status "not stunted" category as a reference in this analysis.

Table 3 shows that mothers with education in the primary school and under category have 1.587 times more likely than mothers with a college education to have stunted children under two years (AOR 1.587; 95% CI 1.576–1.598). Meanwhile, mothers with education in the junior high school category have 1.430 times the probability of mothers with a college education to have stunted children under two years (AOR 1.430; 95% CI 1.420–1.440). Moreover, mothers in the senior high school education category are 1.230 times more likely than mothers with a college education to have stunted children under two years (AOR 1.230; 95% CI 1.222–1.238). This analysis indicates that the lower the level of education, the higher the probability of a mother having stunted children under two years.

In addition to maternal education level, six other independent variables analyzed significantly associated with stunted children under two years. Table 3 informs that mothers who live in urban areas are 0.828 times less likely than mothers who live in rural areas to have stunted children under two years (AOR 0.828; 95% CI 0.825–0.831). The result means that those who live in rural areas have a higher probability of having stunted children under two years.

Based on maternal marital status, mothers who were never married have 1.348 times more likely than divorced/widowed mothers to have stunted children under two years (AOR 1.348; 95% CI 1.308–1.389). Married mothers are 0.804 times less likely than divorced/widowed mothers to have stunted children under two years (AOR 0.804; 95% CI 0.792–0.817). This

**Table 3. Binary logistic regression of nutritional status of children under two years in Indonesia (n = 70,293).**

| Predictors | Stunting | | | |
|---|---|---|---|---|
| | p-value | AOR | 95% CI | |
| | | | Lower Bound | Upper Bound |
| Residence: Urban | < 0.001 | 0.828 | 0.825 | 0.831 |
| Residence: Rural | - | - | - | - |
| Maternal Education: Primary school and under | < 0.001 | 1.587 | 1.576 | 1.598 |
| Maternal Education: Junior high school | < 0.001 | 1.430 | 1.420 | 1.440 |
| Maternal Education: Senior high school | < 0.001 | 1.230 | 1.222 | 1.238 |
| Maternal Education: College | - | - | - | - |
| Maternal age | < 0.001 | 0.994 | 0.994 | 0.995 |
| Maternal Marital Status: Never married | < 0.001 | 1.348 | 1.308 | 1.389 |
| Maternal Marital Status: Married | < 0.001 | 0.804 | 0.792 | 0.817 |
| Maternal Marital Status: Divorced/widowed | - | - | - | - |
| Maternal employment: Unemployed | < 0.001 | 0.972 | 0.968 | 0.975 |
| Maternal employment: Employed | - | - | - | - |
| Children under two years' age | < 0.001 | 1.111 | 1.111 | 1.112 |
| Children under two years' Gender: Boy | < 0.001 | 1.352 | 1.347 | 1.356 |
| Children under two years' Gender: Girl | - | - | - | - |

Note: AOR: Adjusted Odds Ratio; CI: Confidence Interval.

analysis informs that maternal marital status is one predictor of the possibility of children under two years being stunting.

Table 3 indicates that an unemployed mother is 0.972 times less likely than an employed mother to have stunted children under two years (AOR 0.972; 95% CI 0.962–0.975). This information shows that the unemployed mother is a protective factor for having stunted children under two years. Meanwhile, based on age, maternal age, and children's age, the analysis results were significantly associated with the possibility of stunted children under two years.

According to gender, the boy is 1.352 times more likely than the girl to be stunting (AOR 1.352; 95% CI 1.347–1.356). This analysis indicates that children with gender boys have risk factors for experiencing stunting.

## Discussion

We confirmed that the odds of stunting increased significantly among children aged <2 years who had lower maternal education, older (both maternal age and children age), from the rural area, and boys. On the other hand, the study also identified other factors; marital status and occupation. Because of the cross-sectional study design, we cannot exclude the possibility of reverse causation.

This study found lower maternal education is associated with a higher risk of stunting, which agrees with systematic review studies [16–18]. Mothers, as caregivers, have all decisions about healthy feeding practices, including breastfeeding [19, 20]. In addition, higher paternal education was associated with protective caregiving behaviors, including vitamin A capsule receipt, complete childhood immunizations, better sanitation, and the use of iodized salt [21]. We should consider that education is an essential issue for Indonesia, like many other developing countries. Many studies reported a better education level as a strong determinant of better health outcomes [22–25]. Meanwhile, several studies also report poor education as a barrier to achieving better health output [26, 27]. A better level of education can understand the risks and benefits of behavior that will be chosen for adoption [12, 25].

A contradictory finding is older maternal age was associated with a higher risk of stunting. The hypothesis was that younger maternal age could increase more increased risk of stunting. For example, several studies showed the odds of women ≤ 24 years having a stunted child were between 1.09 and 1.23 more significant than women ≥ 33 years [28–30]. Still, some studies have also found contradictory results for maternal age. A previous study found older maternal age has a higher risk of stunting in Indonesia [31]. Older children were significantly associated with a higher risk of stunting in this study, which agrees with a systematic review in Sub-Sahara Africa [18] and a survey among 1366 children aged 0–23 months in Indonesia [32]. We should consider that older children have higher nutrients that are needed. Children who were not given age-appropriate feeding were significantly more likely to be stunted than those fed appropriately [32].

Children who live in rural areas were associated with a higher risk of stunted children under two years. A systematic review concluded that rural residents were associated with stunting [16]. Indonesia's rural health care system was associated with food poverty, low health literacy among parents, mothers' typical household decision-making power, and the consequences of high persistent use of traditional birth attendants among ethnic minorities [33]. In addition, rural subgroups were disadvantaged as the socioeconomic inequalities in maternal, newborn, and child health in Indonesia, i.e., rural people may be insufficient without sufficient skilled local health workers [34]. It suggests that rural area needs more attention for technical and financial support to improve leadership and capacity building in the health sector.

Maternal marital status was associated with stunting in this study. Children with married parents had a lower risk of stunting, and parents who were never married or divorced/widowed had a higher chance of stunting children. In contrast with our study, maternal marital status was not associated with infant growth outcomes in sub-Saharan Africa [35]. Nevertheless, a recent survey in Sub-Sahara Africa indicates that maternal marital status is combined with household cooking fuel on child nutritional status [36].

Meanwhile, employed mothers were one of the risk factors for stunting in children under two years, and employed mothers had a higher risk of stunting children. On the contrary, two studies found no significant association between maternal employment and stunting in Indonesia and Ethiopia [37, 38]. Still, recent studies have also found similar results for maternal marital status in Iran and Ethiopia because housewife mothers have more time to spend with their family and take care of their children [39, 40]. However, maternal marital status is not the leading cause of stunting. Improvements in nutrition-specific and–sensitive sectors, focusing on health care access, sanitation, and education, are critical points to decline in Nepal and Ethiopia [41, 42].

Moreover, boys were more likely to be stunted than girls, and several studies found similar findings in Indonesia, Mozambique, and meta-analysis [37, 43, 44]. We should note that sex and follicle-stimulating hormones might play a role in further growth [45].

## Strength and limitation

The study examines big data to provide results at the national level. Meanwhile, this study analyzed secondary data from the 2017 Indonesia Nutritional Status Monitoring Survey. The variables analyzed were limited to those offered by the survey. The analysis results cannot explain several other variables that have been known from previous studies to affect stunting in children under two years: antenatal care, maternal stature, maternal body mass index, wealth index, diarrhea, anemia, and agri-food [40, 46–48].

On the other side, the study conducted with a quantitative approach cannot explain the associated cultural factors that are still very strong in Indonesia, especially in rural areas. Several previous studies informed the related results, including the value of children, food taboo, parenting, and intake patterns [49–53].

## Conclusions

The study results concluded that maternal education level was associated with stunting among children under two years in Indonesia. The lower the maternal education level, the more likely it is to have stunted children under two years.

Based on the results, the author recommended that the government conduct interventions were focusing on mothers of children under two with poor education to reduce the proportion of stunting under two. A more specific target is mothers of children under five who live in rural areas, are never married, and are employed.

## Author Contributions

**Conceptualization:** Agung Dwi Laksono.

**Data curation:** Ratna Dwi Wulandari, Nurillah Amaliah, Ratih Wirapuspita Wisnuwardani.

**Formal analysis:** Agung Dwi Laksono, Ratna Dwi Wulandari.

**Funding acquisition:** Ratna Dwi Wulandari.

**Investigation:** Ratna Dwi Wulandari, Nurillah Amaliah, Ratih Wirapuspita Wisnuwardani.

**Methodology:** Agung Dwi Laksono.

**Project administration:** Ratna Dwi Wulandari, Ratih Wirapuspita Wisnuwardani.

**Resources:** Ratna Dwi Wulandari, Nurillah Amaliah.

**Software:** Ratna Dwi Wulandari, Nurillah Amaliah, Ratih Wirapuspita Wisnuwardani.

**Supervision:** Agung Dwi Laksono, Ratna Dwi Wulandari.

**Validation:** Agung Dwi Laksono, Ratih Wirapuspita Wisnuwardani.

**Visualization:** Agung Dwi Laksono.

**Writing – original draft:** Agung Dwi Laksono, Ratna Dwi Wulandari, Nurillah Amaliah, Ratih Wirapuspita Wisnuwardani.

**Writing – review & editing:** Agung Dwi Laksono.

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
