## [Decision Letter · Decision Letter 0]

17 Mar 2022

PONE-D-21-21440Stunted among Children Under Two Years in Indonesia: Does maternal education matter?PLOS ONE

Dear Dr. Wulandari,

Thank you for submitting your manuscript to PLOS ONE. After careful consideration, we feel that it has merit but does not fully meet PLOS ONE’s publication criteria as it currently stands. Therefore, we invite you to submit a revised version of the manuscript that addresses the points raised during the review process.

The MS by Laksono and cols addresses an crucial issue in public health, especially in underdeveloping countries, the importance of the parental education and children health. Despite the merit, there are still several points to be cleared as described by the 2 reviewers. Also, an English editing would add quality for this interesting study.  

We look forward to receiving your revised manuscript.

Kind regards,

Marcello Otake Sato, Ph.D., D.V.M.

Academic Editor

PLOS ONE

Journal Requirements:

2. We noticed you have some minor occurrence of overlapping text with the following previous publications, which needs to be addressed:

https://iopscience.iop.org/article/10.1088/1755-1315/755/1/012035

https://onlinelibrary.wiley.com/doi/epdf/10.1111/mcn.12826

http://mji.ui.ac.id/journal/index.php/mji/article/view/1819

https://worldwidescience.org/topicpages/c/canada+temporal+socio-demographic.html

The text that needs to be addressed involves the Introduction.

In your revision ensure you cite all your sources (including your own works), and quote or rephrase any duplicated text outside the methods section. Further consideration is dependent on these concerns being addressed.

3. Please address the following:

- Please refrain from stating p values as 0.000 and instead use the format p<0.0001.

- Please re-read the Abstract of this manuscript and ensure it contains no errors of grammar. 

- Please ensure that you have discussed whether all data were fully anonymized before you accessed these data.

No

6. We note that Figure 1 in your submission contain [map/satellite] images which may be copyrighted. All PLOS content is published under the Creative Commons Attribution License (CC BY 4.0), which means that the manuscript, images, and Supporting Information files will be freely available online, and any third party is permitted to access, download, copy, distribute, and use these materials in any way, even commercially, with proper attribution. For these reasons, we cannot publish previously copyrighted maps or satellite images created using proprietary data, such as Google software (Google Maps, Street View, and Earth). For more information, see our copyright guidelines: http://journals.plos.org/plosone/s/licenses-and-copyright.

Reviewers' comments:

Reviewer's Responses to Questions

**Comments to the Author**

1. Is the manuscript technically sound, and do the data support the conclusions?

Reviewer #1: Yes

Reviewer #2: Yes

2. Has the statistical analysis been performed appropriately and rigorously? 

Reviewer #1: Yes

Reviewer #2: Yes

3. Have the authors made all data underlying the findings in their manuscript fully available?

Reviewer #1: Yes

Reviewer #2: Yes

4. Is the manuscript presented in an intelligible fashion and written in standard English?

Reviewer #1: Yes

Reviewer #2: Yes

5. Review Comments to the Author

Reviewer #1: This study addresses an important issue in global public health problem, i.e., stunting, in Indonesia, one of the largest countries in the world.

Title:

• As a non-native speaker, I am not sure about using the word “stunted among…..”. Please check.

Abstract:

• Authors could include the p-value or 95%CI for the point estimates listed.

• Authors might want to conclude the results more rather than restating the results.

Introduction:

• Authors might want to revisit the first sentence regarding the definition of stunting based on the WHO.

• The Introduction Section could be made more concise.

Materials and methods

• Please add a brief explanation about the 2017 Indonesian Nutritional Status Monitoring.

• Were there any variables related to household economic status apart from maternal employment? The economic status of mothers or households is an important indicator to adjust in the model, and this might be related to maternal education.

• Did the authors consider using province/region to adjust in the analysis?

• It will be good to explain how the survey conducted anthropometric measurement. Please elaborate.

• Were there any extreme values of these measurements found? How were they treated?

Results:

• Please add the 95%CI for the proportion of stunted children reported.

• The authors mentioned the collinearity test between dependent and independent variables? Please clarify.

• As a non-native speaker, I am not familiar with the wordings used by authors to describe Table 2 – “occupy” or “controlled” or “rule,” so I could not make any advice about this.

• Table 3. Please change p=0.000 to p>0.001, and authors can remove the Asterix for the p-value in the footnote.

• Table 3. The lower and upper bound for aOR for maternal age is the same? Please check again.

• Table 3. To make it easier for readers, authors could change some reference groups to make it more consistent, e.g., selecting ‘employed’ as the reference group, instead of ‘unemployed,’ so that all aOR consistently indicates the risk factors rather than a mixed with the protective factor (<1.0).

Discussion:

• Please avoid using the word “influence” or “effect.” It is best to use “association” as this is a cross-sectional study.

• Overall, the authors might include some recommendations of interventions that could be implemented based on the findings. Authors could use references from previous studies/literature regarding the effectiveness of those interventions.

• Authors might want to add the strengths of this study.

Conclusions:

• Authors might want to conclude the findings rather than restating the findings. Authors could include some recommendations based on the findings.

Reviewer #2: - I recommend revision of the title indicating that this is a meta-analysis and use the word "stunting" or "stunted growth" instead of stunted in the title.

- Abstract : Background - Expound a bit on the background of the study; use "stunting" among children...

Methods - What is maternal marital? Does it pertain to maternal status?

- Introduction - Avoid very long sentences and repeated use of "and" in a sentence (last sentence, second paragraph)

- Materials and Methods - Again, use stunting (a condition, status) and not stunted; You may include the inclusion and exclusion criteria and other statistical analysis/es employed in the study

-Results - Please follow proper format for Tables (APA format); format Tables 1-3

-Table 1 - Since maternal characteristics is written in bold and clear, I suggest to delete the "mother's.." in the 4 entries

-Conclusions - Use "stunting among children under two years in Indonesia"

6. PLOS authors have the option to publish the peer review history of their article (what does this mean?). If published, this will include your full peer review and any attached files.

Reviewer #1: No

Reviewer #2: No

---

## [Author Response · Author response to Decision Letter 0]

18 May 2022

Response

1. We note that Figure 1 in your submission contain [map/satellite] images which

may be copyrighted. All PLOS content is published under the Creative

Commons Attribution License (CC BY 4.0), which means that the manuscript,

images, and Supporting Information files will be freely available online, and

any third party is permitted to access, download, copy, distribute, and use

these materials in any way, even commercially, with proper attribution. For

these reasons, we cannot publish previously copyrighted maps or satellite

images created using proprietary data, such as Google software (Google

Maps, Street View, and Earth). For more information, see our copyright

guidelines: http://journals.plos.org/plosone/s/licenses-and-copyright.

We require you to either (1) present written permission from the copyright

holder to publish these figures specifically under the CC BY 4.0 license, or (2)

remove the figures from your submission:

a. You may seek permission from the original copyright holder of Figure 1 to

publish the content specifically under the CC BY 4.0 license.

We recommend that you contact the original copyright holder with the Content

Permission Form (http://journals.plos.org/plosone/s/file?id=7c09/contentpermission-

form.pdf) and the following text:

“I request permission for the open-access journal PLOS ONE to publish XXX

under the Creative Commons Attribution License (CCAL) CC BY 4.0

(http://creativecommons.org/licenses/by/4.0/). Please be aware that this

license allows unrestricted use and distribution, even commercially, by third

parties. Please reply and provide explicit written permission to publish XXX

under a CC BY license and complete the attached form.”

Please upload the completed Content Permission Form or other proof of

granted permissions as an "Other" file with your submission.

In the figure caption of the copyrighted figure, please include the following

text: “Reprinted from [ref] under a CC BY license, with permission from [name

of publisher], original copyright [original copyright year].”

b. If you are unable to obtain permission from the original copyright holder to

publish these figures under the CC BY 4.0 license or if the copyright holder’s

requirements are incompatible with the CC BY 4.0 license, please either i)

remove the figure or ii) supply a replacement figure that complies with the CC

BY 4.0 license. Please check copyright information on all replacement figures

and update the figure caption with source information. If applicable, please

specify in the figure caption text when a figure is similar but not identical to the

original image and is therefore for illustrative purposes only.

The authors decided to remove the figure from the manuscript.

---

## [Decision Letter · Decision Letter 1]

4 Jul 2022

Stunting among Children Under Two Years in Indonesia: Does maternal education matter?

PONE-D-21-21440R1

Dear Dr. Wulandari,

We’re pleased to inform you that your manuscript has been judged scientifically suitable for publication and will be formally accepted for publication once it meets all outstanding technical requirements.

Kind regards,

Marcello Otake Sato, Ph.D., D.V.M.

Academic Editor

PLOS ONE

Additional Editor Comments (optional):

Should the authors correct these points at the proofreading step:

-Abstract: Methods- 3rd statement - maternal marital - It should be written as maternal marital status

-Results: Last sentence- Should be: Moreover...mothers with a college education to have stunted children under two years (...).

-Conclusion: Last sentence should be written: The lower the mother's... chances of a mother having stunted children under two years.

-Page 6: Last line- Multivariate Analysis: Under two years of stunted growth (...)

-Page 7: First statement- Should be written as ...mothers having stunted children under two years.

-Second paragraph - same comment as previous.

-NOTE: Please check all phrases/statements written as - ...having children under two years stunted... should be written as ...having stunted children under two years. The way the statement was written seems not to convey sense of meaning. Kindly change to: ...having stunted children under two years.

-Conclusion: First paragraph - the lower the maternal... to have stunted children under two years. OR ...to have stunted children under two years of age.

Reviewers' comments:

Reviewer's Responses to Questions

**Comments to the Author**

1. If the authors have adequately addressed your comments raised in a previous round of review and you feel that this manuscript is now acceptable for publication, you may indicate that here to bypass the “Comments to the Author” section, enter your conflict of interest statement in the “Confidential to Editor” section, and submit your "Accept" recommendation.

Reviewer #2: All comments have been addressed

2. Is the manuscript technically sound, and do the data support the conclusions?

Reviewer #2: Yes

3. Has the statistical analysis been performed appropriately and rigorously? 

Reviewer #2: Yes

4. Have the authors made all data underlying the findings in their manuscript fully available?

Reviewer #2: Yes

5. Is the manuscript presented in an intelligible fashion and written in standard English?

Reviewer #2: No

6. Review Comments to the Author

Reviewer #2: Abstract: Methods- 3rd statement - maternal marital - It should be written as maternal marital status

Results: Last sentence- Should be: Moreover...mothers with a college education to have stunted children under two years (...).

Conclusion: Last sentence should be written: The lower the mother's... chances of a mother having stunted children under two years.

Page 6: Last line- Multivariate Analysis: Under two years of stunted growth (...)

Page 7: First statement- Should be written as ...mothers having stunted children under two years.

Second paragraph - same comment as previous.

NOTE: Please check all phrases/statements written as - ...having children under two years stunted... should be written as ...having stunted children under two years. The way the statement was written seems not to convey sense of meaning. Kindly change to: ...having stunted children under two years.

Conclusion: First paragraph - the lower the maternal... to have stunted children under two years. OR ...to have stunted children under two years of age.

7. PLOS authors have the option to publish the peer review history of their article (what does this mean?). If published, this will include your full peer review and any attached files.

Reviewer #2: No

---

## [Editor Report · Acceptance letter]

15 Jul 2022

PONE-D-21-21440R1 

Stunting among children under two years in Indonesia: Does maternal education matter? 

Dear Dr. Wulandari:

I'm pleased to inform you that your manuscript has been deemed suitable for publication in PLOS ONE. Congratulations! Your manuscript is now with our production department. 

Kind regards, 

on behalf of

Dr. Marcello Otake Sato 

Academic Editor

PLOS ONE